# *Babesia gibsoni* Infection in Dogs—A European Perspective

**DOI:** 10.3390/ani12060730

**Published:** 2022-03-14

**Authors:** Oliwier Teodorowski, Marcin Kalinowski, Dagmara Winiarczyk, Banu Dokuzeylül, Stanisław Winiarczyk, Łukasz Adaszek

**Affiliations:** 1Veterinary Clinic “Teodorowscy”, 43-190 Mikołów, Poland; oliwierteodorowski@polvet.pl; 2Department of Epizootiology and Clinic of Infectious Diseases, Faculty of Veterinary Medicine, University of Life Sciences, 20-612 Lublin, Poland; genp53@interia.pl (S.W.); lukasz.adaszek@up.lublin.pl (Ł.A.); 3Department and Clinic of Animal Internal Diseases, Faculty of Veterinary Medicine, University of Life Sciences, 20-612 Lublin, Poland; winiarczykdm@gmail.com; 4Department of Internal Medicine, Faculty of Veterinary Medicine, Istanbul University—Cerrahpasa, Istanbul 34320, Turkey; bdokuzeylul@gmail.com

**Keywords:** *Babesia gibsoni*, babesiosis, dog, Europe

## Abstract

**Simple Summary:**

Canine babesiosis is a tick-borne, protozoal, hemoparasitic disease caused by infection by parasites of the genus *Babesia*. Numerous species of *Babesia* exist worldwide. These protozoa are classified as either large forms (e.g., *Babesia canis*) or small forms (e.g., *Babesia gibsoni*). Reports of infections with small protozoan species are far less numerous. In most European countries where *B. gibsoni* has been reported, the percentage of infected dogs is around 1%. The literature review suggests that *B. gibsoni* should not only be considered a random and imported pathogen, but also a possible emerging parasite in Europe. The disease is much more severe than *B. canis* infections in most cases. Accurate molecular detection and species identification are important for selecting the correct therapy and predicting the course of the disease in dogs with babesiosis. In the future, it is expected that *B. gibsoni* infections may appear in other non-endemic regions in Europe, which may pose significant diagnostic and therapeutic challenges for veterinary practitioners.

**Abstract:**

Canine babesiosis is a disease caused by infection with parasites of the genus *Babesia*. These protozoa are classified as either large (e.g., *Babesia canis*) or small (e.g., *Babesia gibsoni*). So far, only three small *Babesia* species of clinical importance, able to infect dogs, have been described: *B. gibsoni*, *B. conradae*, and *B. vulpes*. This review presents the current epidemiological situation of *Babesia gibsoni* infections in dogs in Europe. In most European countries where *B. gibsoni* has been reported, the percentage of infected dogs is around 1%. The higher prevalence of the *B. gibsoni* infection among American Pit Bull Terriers suggests breed susceptibility. An analysis of the available data makes it possible to conclude that *B. gibsoni* infections may appear in the future in other non-endemic regions of Europe, which may pose significant diagnostic and therapeutic challenges for veterinary practitioners.

## 1. Introduction

Canine babesiosis is a disease caused by infection with parasites of the genus *Babesia* [1]. Numerous species of *Babesia* exist worldwide. Traditionally, the morphology of the protozoan (piroplasm merozoites) within red blood cells was used as the main taxonomic determinant. These protozoa are classified as either large forms (e.g., *Babesia canis*) or small forms (e.g., *Babesia gibsoni*) (Figure 1). Subsequently, molecular techniques have allowed the identification and differentiation of several species of *Babesia* that can infect dogs [2,3]. Tóthová et al. [4] also demonstrated that, based on the protein profile of the serum of dogs infected with protozoans, it is possible to determine which of the protozoans (large or small forms) was the cause of infection. These authors determined that in the serum of dogs infected with *B. gibsoni*, a higher concentration of gamma-globulins and a markedly lower albumin concentration was observed than in dogs infected with *B. canis*.

The main *Babesia* species that infect dogs in Europe are *B. canis* and *B. vogeli* [2,5,6,7]. Reports of infections with small protozoan species, when compared to *B. canis*, are far less numerous. So far, only three small *Babesia* species with clinical importance have been described: *B. gibsoni*, *B. conradae* [8], and the recently reported *B. vulpes* [9].

Clinical manifestations of *B. gibsoni* infections resemble *B. canis* (lymphadenopathy, enlargement of the spleen, diarrhea, weight loss, proteinuria, hemoglobinuria, polyuria/polydipsia (PU/PD), nephropathy). The course of the disease in most cases is severe [10,11,12]. Interestingly, breeds such as the Tosa Inu and American Staffordshire Terrier tend to be predisposed to the development of the disease, where it tends to be more severe than in other breeds. This may explain the difficulty in treating these breeds [11]. Therefore, it seems important to be aware of the prevalence of *B. gibsoni* in Europe to consider infections with these parasites in the differential diagnosis of vector-borne diseases—especially disorders involving thrombocytopenia.

## 2. Pathogenesis

The infections of dogs occur during infestations with arachnids when the blood-sucking ticks introduce sporozoites into the dog’s body [13,14,15]. Known vectors of *B. gibsoni* are hard ticks of the genus *Rhipicephalus* (*R. sanguineus*) and *Haemphysalis* (*H. longicornis*) [14,16].

Transmission routes of lesser importance are blood transfusion and contact between open wounds and infected blood. It is likely that the invasion can also spread from the mother to the offspring via the placenta. Stegman et al. [17] describe the infection of a female German Shepherd with *Babesia* protozoa via blood transfusion. Wound contact with infected blood may occur during dog fights. Not surprisingly, the breeds most frequently infected this way are the American Pit Bull Terrier and the Tosa Inu, especially in the United States and Asia, where dog-fighting—although illegal—is very popular [18,19].

Canine babesiosis may follow various courses depending on the parasite strain that causes the disease, the age of the animal, its immunological status, breed, origin, and co-morbidities. Uncomplicated babesiosis is characterized by anemia [20,21], while a complicated form is characterized by organ dysfunction [20,22,23].

Canine babesiosis involves extravascular and intravascular hemolysis, manifested by regenerative anemia, hemoglobinemia, hemoglobinuria and bilirubinuria. Multiple mechanisms play a role in the development of hemolytic anemia in the course of the *B. gibsoni* infection. The disintegration of erythrocytes may be caused by their mechanical damage due to replicating parasites, as well as through their damage by antibodies, the complement system, and oxidizing factors [20,24,25,26].

*B. gibsoni*-infected dogs with parasitemia showed a significant increase of both methemoglobin and malondialdehyde (an end product of lipid peroxidation) concentration in erythrocytes compared to those in uninfected dogs. Furthermore, erythrocytes from parasitized culture were more susceptible to phagocytosis by bone marrow macrophages from uninfected dogs than erythrocytes from the control culture. In addition, macrophages ingested not only parasitized erythrocytes but also non-parasitized cells. These results suggested that oxidative damage to erythrocytes was induced by the multiplications of *B. gibsoni*, and that non-parasitized erythrocytes were also exposed to erythrocytes oxidative stress during the infection by *B. gibsoni*, which increases anemia [26,27].

Similar to the *B. canis* infection, a probable consequence of anemia in the course of the *B. gibsoni* infection may be the development of hypoxia and a rightward shift of the hemoglobin dissociation curve (the reduced ability of hemoglobin to carry oxygen), resulting in a gradual increase in the carboxyhemoglobin fraction [28,29].

The variety of clinical symptoms encountered in the course of canine babesiosis due to the *B. gibsoni* infection is not due to the direct impact of parasites on tissues and organs, but rather the inflammatory processes induced by their presence—which affect different organs and systems to varying degrees. The main mediators of these inflammatory reactions are cytokines, nitric oxide, and free oxygen radicals. Cytokines, which are responsible for mediating and regulating all aspects of the immune response to infection, play an important role in inducing systemic inflammation [30]. The only cytokine identified, but associated with the *B. gibsoni* infection, is tumor necrosis factor-alpha (TNF-α), which was found in higher concentrations in dogs with higher peripheral parasitemia and a more severe disease [31].

Although the disease can manifest itself in different ways, the pathomechanism for developing clinical symptoms is universal and underlies the disorders described above [32].

## 3. Distribution of *Babesia gibsoni* in European Countries

The geographical distribution of *Babesia* spp. depends on the presence of competent ticks to transmit each of them. Thus far, not all such species have been identified in Europe (Table 1). Confirmed cases of *B. gibsoni* infections in dogs have been reported in only a few European countries (Table 1).

In Europe, the vector of *B. gibsoni* is *Rhipicephalus sanguineus* [33], although it should be remembered that infections with these protozoa are not only transmitted by arachnids, as the infection may be transmitted among fighting dog breeds independently of the limitations of the tick infestation vector [34] or after blood transfusions [17].

The first case of the *B. gibsoni* infection in a dog in Europe was in 2003 in Spain [35]. The disease was diagnosed in a dog that exhibited typical symptoms of babesiosis (apathy, sadness, jaundice, hemoglobinuria). Molecular biology methods confirmed the infection and the 18S rRNA gene sequences obtained showed 98% homology with a *B. gibsoni* isolate from a dog in Malaysia (isolate Asia 1).

In 2009, Tabar et al. [36] carried out molecular monitoring of dogs from the Barcelona area for *Leishmania infantum* and *Ehrlichia*, *Anaplasma*, *Rickettsia*, *Bartonella*, *Hepatozoon*, *Babesia*, and *Theileria*. The test material consisted of blood samples collected from 153 dogs. The presence of *B. gibsoni* DNA was found in 2% of the tested samples, which confirmed the presence of *B. gibsoni* in Spain.

Although there were already reports of small piroplasm invasions in dogs in Spain, Suarez et al. [37] published a case of a fatal *B. gibsoni* infection in a Spanish dog. No genetic studies were carried out and, thus, the final identification of the parasite was unclear.

Germany is another European country where dogs infected with *B. gibsoni* have been found. In 2007, Hartelt et al. [38] described the first two autochthonous cases of *B. gibsoni* (Asian genotype) infection. The disease was diagnosed in two American Pit Bull Terriers which that exhibited typical symptoms of babesiosis. Protozoan DNA extracted from the blood of the diseased animals showed 100% nucleotide sequence homology of the 18S RNA gene to the Asian genotype of *B. gibsoni*. On the other hand, Hamel et al. [3] and Schäfer et al. [39] showed that the growing problem related to dog infections with *B. gibsoni* in Germany mainly stems from the recently intensified importation of dogs from areas of endemic distribution of tick-borne diseases, including countries in the Mediterranean region and south-eastern Europe (Romania and Hungary).

In subsequent years, an increase in the range of *B. gibsoni* was observed in southern and eastern Europe. In Croatia, the presence of *B. gibsoni* DNA was demonstrated in the blood of 6 out of 848 randomly selected, asymptomatic dogs (Asian genotype, GenBank Accession Nos. AF175300, AF175301, AB118032, AF205636, AF271081, AF271082, and AY278443) [40]. The six isolates of *B. gibsoni* were identical to isolates from Japan, the United States, Australia, Spain, and Germany. The blood samples in which the presence of genetic material of protozoa was detected came from American Pit Bull Terriers all living in a village close to the Bosnian border. Surprisingly, these animals did not show any signs of disease although it is generally accepted that *B. gibsoni* infections in dogs are severe and that those reported to date in dogs in Europe have manifested acute clinical signs of babesiosis. Asymptomatic infections with this species have certainly been reported [21,41]; nevertheless, detecting the pathogenic species of *B. gibsoni* in asymptomatic dogs indicates the relationship between parasite species/subspecies and clinical signs of infection in dogs deserves further investigation.

In the same year, Trotta et al. [42] described the first clinical case of babesiosis due to *B. gibsoni* in a dog in Italy. The disease was diagnosed in a four-year-old female American Pit Bull Terrier that exhibited symptoms of weakness, a lack of appetite, vomiting, splenomegaly, pale mucous membranes, and abdominal pain. The presence of small protozoa in the erythrocytes was confirmed by microscopic examination of blood smears, while the final diagnosis of the disease was based on the results of molecular testing and sequencing of the obtained genetic material. As the dog was born in Croatia from a bitch imported from the USA and transferred to Italy at the age of four months, intrauterine transmission is suspected [22,43].

In the following years, Veneziano et al. [44] carried out serological monitoring for babesiosis among 1311 healthy hunting dogs from 153 municipalities located in southern Italy. The overall B. *gibsoni* seroprevalence was 0.2% (3/1311), although *B. gibsoni* DNA was not amplified using qPCR. The low *B. gibsoni* seroprevalence and absence of any *B. gibsoni* PCR-positive animal among a large number of hunting dogs may suggest that tick transmission of *B. gibsoni* does not occur in southern Italy. Molecular monitoring of blood samples from 607 dogs collected between January 2016 and December 2019 and submitted to San Marco veterinary laboratory showed the presence of *B. gibsoni* DNA in only six of them (0.99%). *B. gibsoni* was mainly isolated from male (6/6, 100%), purebred (5/6, 83.3%) dogs with a median age of 40.5 months [45].

A relatively high prevalence of the *B. gibsoni* infection has been found in dogs in Romania [46]. In a study involving 49 dogs with typical symptoms of babesiosis from five western and north-western counties of Romania, *B. gibsoni* was identified and differentiated by PCR-RFLP targeting the 18S rRNA gene in 14 of 49 blood samples (28.6%). All of the *B. gibsoni* sequences showed ≥99% homology with the 18S rRNA *B. gibsoni* isolates deposited in GenBank from Japan (AB478330), Taiwan (FJ769388), and the United States (AF396749). Of the 14 dogs examined, 13 belonged to fighting dog breeds (12 American Pit Bull Terriers and one American Staffordshire Terrier). Six dogs never traveled with their owners or left their homes, which may indicate that, in their case, the *B. gibsoni* infection was considered autochthonous in origin. The remaining eight dogs originated from Hungary or had traveled to Hungary with their owners, which may indicate that, in the case of these dogs, babesiosis had entered Romania from this country.

Andersson et al. [47] also conducted molecular monitoring for protozoan diseases in the dog population in Romania. The study included 96 symptomatic dogs, of which the presence of *B. gibsoni* DNA was demonstrated in only one animal (i.e., corresponding to a prevalence of 1%). In contrast to the results obtained by Imre et al. [46], the obtained protozoan nucleotide sequence (KY433318) was identical to a sequence from Slovakia, GenBank accession number KP737862. By analyzing the prevalence of *B. gibsoni* in dogs in Romania, it can be concluded that there are differences in the geographical spread of the protozoa within the country. Dogs originating from the western and north-western regions of Romania appear to be more vulnerable to infection with these parasites than the animals which originating from the south of the country [46,47].

Another European country where *B. gibsoni* was reported in dogs was Serbia. During the years 2012–2014, a total of 158 outdoor dogs from Pančevo and Durdevo (northern Serbia) and Niš and Prokuplje (southern Serbia) were submitted for molecular analyses concerning canine babesiosis. The overall prevalence of *B. gibsoni* was 5.7% [48]. In contrast, in dogs from Belgrade, the molecular prevalence of the *B. gibsoni* infection was 2.7%, while the seroreactivity for these piroplasms was 12.6% [49]. No significant difference between infected and noninfected dogs was found by age, sex, or place of residence, whereas there were differences regarding the presence of ticks and the application of preventive measures [48]. Two other *B. gibsoni* infections in Tosa Inu and American Staffordshire Terrier dogs in Serbia were described by Davitkov et al. [50]. The 18S rDNA partial sequences of the parasites showed the highest similarity (100% correspondence) with sequences of *B. gibsoni* obtained from naturally infected dogs in India (GenBank accession no. KF878947.1), China (HG328237.1), Saint Kitts and Nevis (JX112784.1), Japan (AB478330.1), Taiwan (FJ769388.1), the USA (EU084677.1), Australia (AY102164.1), and Spain (AY278443.1).

The first cases of the *B. gibsoni* infection in dogs in eastern Europe occurred in Slovakia [51] and Poland [52]. Vichova et al. [51] described cases of the disease in two mixed pit bull dogs aged three (Dog 1) and two years old (Dog 2). Dog 1 was taken to a veterinary clinic in Bratislava with a history of massive hemoglobinuria, while Dog 2 had a three-day history of lethargy, weakness, and repeated blackouts. The blood of both dogs showed the presence of *B. gibsoni* DNA. The sequence of the amplified DNA fragment showed 100% similarity to the sequences with *B. gibsoni* isolated from dogs from Japan (LC012808), Australia (AY102164), and Spain (AY278443). In both cases, the parasites showed resistance to imidocarb, and only treatment with Malarone (atovaquone and proguanil hydrochloride) (13.3 mg/kg once per 8 h) and azithromycin (10 mg/kg once per day) per os (PO) for ten days proved effective.

Adaszek et al. [52], on the other hand, described the first case of the *B. gibsoni* infection in Poland in a four-year-old American Staffordshire Terrier. The dog was brought to the veterinary clinic with symptoms of severe weakness, apathy, and hematuria. The owners did not report the presence of ticks, nor had they recently traveled abroad with the animal. The blood analysis indicated the presence of *B. gibsoni* DNA, and the sequence analysis of the 18S rRNA gene of the protozoa also indicated the presence of the *B. gibsoni* LC012808.1 sequence at about 98%. Treating the dog with diminazene 5 mg/kg (once daily), metronidazole 25 mg/kg every 8–12 h for three weeks, clindamycin 12.5 mg/kg twice daily for three weeks led to a significant improvement in the health of the dog.

In the following years, also in Poland, Teodorowski et al. [7] carried out molecular monitoring for vector-borne diseases of the blood samples of 216 dogs. The authors detected *B. gibsoni* DNA in three of them, which showed high homology (99.8–100%) to the sequence *B. gibsoni* LC012808.1. This was also shown in an earlier study by Adaszek et al. [52].

**Table 1 animals-12-00730-t001:** Occurrence of canine infection by *Babesia gibsoni* in Europe based on molecular analysis results.

Country/Region	No. of Examined Dogs	No. of Infected Dogs (%)	Technique	Comments	Reference
Spain	1	1 *	PCR	Symptomatic infection	[35]
Spain (Barcelona)	153	3 (2)	PCR	Symptomatic, and asymptomatic infection	[36]
Germany	2	2 *	PCR	Symptomatic infection, in American Pit Bull Terriers	[38]
Croatia	929	6 (0.6)	PCR	Asymptomatic infection, in fighting dogs	[40]
Italy	607	6 (1)	PCR	Symptomatic, and asymptomatic infection	[45]
Romania	49	14 (28.6)	PCR	Symptomatic infection, in fighting dog breeds	[46]
	96	1 (1)			[47]
Serbia	9	158 (5.7)	PCR	Asymptomatic infection	[48]
	3	111 (2.7)			[49]
Slovakia	17	17 **	PCR	Symptomatic infection, in mixed pit bull terriers and American Staffordshire Terriers	[4]
	2	2 *			[51]
Poland	216	3 (1.4)	PCR	Symptomatic infection, in American Staffordshire Terriers	[7]
	1	1 *			[52]

Legend: * case reports; ** research study performed on 17 *B. gibsoni* infected dogs.

The literature review presented here indicates that *B. gibsoni* infections in dogs in Europe are not frequent and are far less frequent than *B. canis* infections. Among 44 European countries, the *B. gibsoni* infections in dogs were described in only eight of them (Figure 2). In most countries where babesiosis has been reported against a background of the small piroplasms discussed, the percentage of infected dogs was approximately 1% of those tested (1%, 6/607 in Italy [45]; 0.6%, 6/929 in Croatia [40]; 1.4%, 3/216 in Poland [7]; and 1%, 1/96 in Romania [47]).

A higher prevalence was only recorded in western and north-western Romania (28.6%) [46] and Serbia (2.7–5.7%) [48,49]. Singular cases were described in Spain [35], Germany [38], and Slovakia [51].

In several studies, cases of infection were reported in American Pit Bull Terriers and related breeds [38,40,42,46,51,52]. The obviously larger number of cases in these dog breeds could be explained by breed-specific susceptibility to *B. gibsoni* or, alternatively, by transmission of this pathogen via wounds, saliva, or ingested blood as a result of bite injuries [19].

The literature review suggests that *B. gibsoni* should not only be considered as a random and imported pathogen, but rather a possible emerging parasite in Europe. This has important implications for veterinary practice.

Blood smear observation should be the “first-step” diagnostic tool. In cases of suspected babesiosis and a negative finding in the blood smear, it is appropriate to analyze a blood sample using the PCR technique or indirect fluorescent antibody test (IFAT) [53,54]. To identify the species of piroplasm, morphological observation is insufficient, and molecular techniques such as PCR and sequencing are necessary [54,55].

*B. gibsoni* is resistant to traditional antibabesial therapy with imidocarb dipropionate, which is effective against *B. canis*. Similarly, the effect of diminazene aceturate (used for the treatment of equine or bovine babesiosis) is limited in the control of *B. gibsoni* infections [55,56]. The effectiveness of imidocarb dipropionate or diminazene aceturate treatment of *B. gibsoni* infections mostly lies in the reduction in the severity of the clinical signs [57]. Treatment is problematic and often accompanied by frequent clinical relapses [58,59]. Many antibiotics have been tried in the treatment of *B. gibsoni* infections. Some studies describe the administration of antibiotics as monotherapy; for example, clindamycin [57] or enrofloxacin (in vitro experiment) [60]. Their therapeutic use had the effect of reducing and partially adjusting the blood parameters, but there was no absolute cure in most cases.

As a more effective therapy, a combination of the antimalarial atovaquone (naphthoquinone) and the macrolide azithromycin, or a combination of clindamycin, metronidazole and doxycycline is recommended. This ensures the elimination of infection and leads to significant suppression of parasitemia [11,61].

## 4. Conclusions

In conclusion, this infection should be suspected if the patient is an American Pit Bull Terrier or a related dog breed. In the future, it is expected that *B. gibsoni* infections may appear in other non-endemic regions in Europe, which may pose significant challenges for veterinary practitioners. Constant monitoring of the *B. gibsoni* infection cases in dogs in Europe, as well as the creation of maps of their occurrence in individual countries (Figure 2) will allow for an easier analysis of the epidemiological situation of this tick-borne disease. It will contribute to its more effective diagnosis, therapy, and prevention.

## Figures and Tables

**Figure 1 animals-12-00730-f001:**
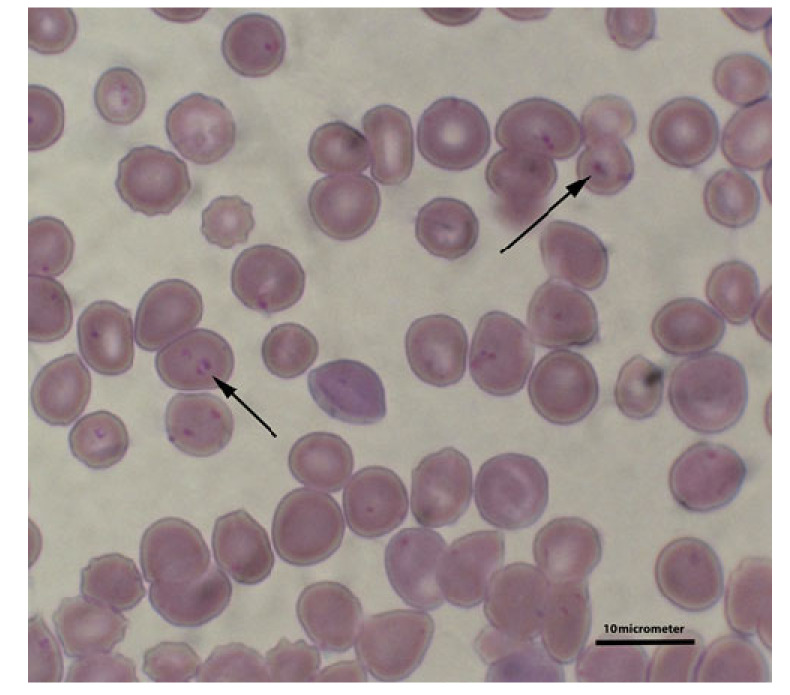
*B. gibsoni* inside red blood cells (marked with arrows). Blood smear stained by the Giemsa method.

**Figure 2 animals-12-00730-f002:**
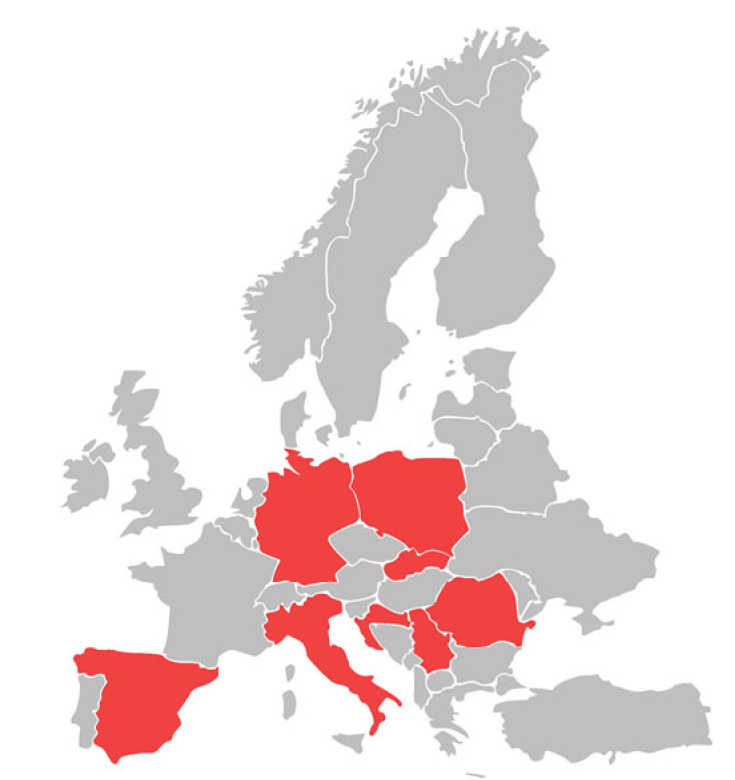
Occurrence of canine infection by *Babesia gibsoni* in European countries.

## Data Availability

This article is a literature review, therefore, all data described here were obtained from the works cited in the references.

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
