# Peer review of "Babesia gibsoni Infection in Dogs—A European Perspective"

_animals, 2022, doi:10.3390/ani12060730_

Round 1

Reviewer 1 Report

Review of the MS No. animals-1579344 entitled "Babesia gibsoni Infection in Dogs – a European Perspective"

This is review article presenting current knowledge about Babesia gibsoni infections in Europe. The manuscript is well written and interesting. References is the main problem with this article. There is high contrast in citations between chapters 2 and 3. Chapter 2 requires proper citations, especially based on original works, not review articles. Moreover, some of mechanisms in pathogenesis presented in chapter 2 were observed in dogs infected with large Babesia, mainly B. canis and B. rossi. Thus the authors should emphasize this using proper citations, and speculate that probably similar mechanisms may occur in dogs infected with small Babesia.

Specific comments:

Lines 52-53: B. rossi is not one of the main species of the genus Babesia infecting dogs in Europe.

Line 56 (Table 1):

  • Lack of citations.
  • The name of Babesia microti-like is Babesia vulpes.
  • rossi is transmitted by Haemaphysalis elliptica.
  • Lack of Theileria equi which was also detected in dogs.

Line 59: I think it's better to write haemoglobinuria instead of haematuria.

Line 64: Redundant sentence. It was mentioned earlier that B. gibsoni causes more severe infection than B. canis.

Lines 69-71: Lack of citations.

Lines 81-85: These complications described in article No. 16 were detected in dogs infected with large Babesia. The authors of the article No. 16 have written "Multiple organ dysfunction syndrome is documented in canine babesiosis caused by B. rossi and B. canis". Thus the authors should mention about this, and speculate if MODS develop in B. gibsoni infection.

Lines 87-104: The mechanism presented here is not described in the cited article No. 16 at the end of the paragraph.

Lines 105-123: The whole paragraph (2.1. Sequestration - by the way I think it should be 2.2.) is based on one citation. Can the authors cite any studies on B. gibsoni associated with the mechanism described in this paragraph? E.g. One study showed increased concnetration of TNF in B. gibsoni infected dogs (https://doi.org/10.1016/j.exppara.2015.08.002)

Line 126: I think the anemia is regenerative. No word about degenerative anemia in cited article No. 18.

Lines 131-136: This mechanism has not been described in cited here article No. 19.

Lines 139-146: I cannot verify this paragraph. I have no access to the article No. 20.

Author Response

AUTHORS’ RESPONSES TO REVIEWER #1

We are very grateful to the Reviewer for his/her careful reading of the manuscript.

This is review article presenting current knowledge about Babesia gibsoni infections in Europe. The manuscript is well written and interesting. References is the main problem with this article. There is high contrast in citations between chapters 2 and 3. Chapter 2 requires proper citations, especially based on original works, not review articles. Moreover, some of mechanisms in pathogenesis presented in chapter 2 were observed in dogs infected with large Babesia, mainly B. canis and B. rossi. Thus the authors should emphasize this using proper citations, and speculate that probably similar mechanisms may occur in dogs infected with small Babesia.

We are very grateful to the Reviewer for his/her careful reading of the manuscript. We agree with all suggestion of the reviewer. The manuscript was reformatted. Section 2.1, and 2.2 were removed from the text. In a present form information presented in the text are focused mainly on B. gibsoni infection as suggested by the Reviewer. Proper citations were used in the text.

Specific comments:

Lines 52-53: B. rossi is not one of the main species of the genus Babesia infecting dogs in Europe.

That is right. This information was removed from the text.

Line 56 (Table 1): Lack of citations. The name of Babesia microti-like is Babesia vulpes. rossi is transmitted by Haemaphysalis elliptica. Lack of Theileria equi which was also detected in dogs.

According to the suggestion of the Editor, Table 1 was removed from the text.

Line 59: I think it's better to write haemoglobinuria instead of haematuria.

Text was changed as suggested by the Reviewer.

Line 64: Redundant sentence. It was mentioned earlier that B. gibsoni causes more severe infection than B. canis.

The sentence was removed from the text, as suggested by the Reviewer.

Lines 69-71: Lack of citations.

The citations were added to the text as suggested by the Reviewer.

Lines 81-85: These complications described in article No. 16 were detected in dogs infected with large Babesia. The authors of the article No. 16 have written "Multiple organ dysfunction syndrome is documented in canine babesiosis caused by B. rossi and B. canis". Thus the authors should mention about this, and speculate if MODS develop in B. gibsoni infection.

The “Pathogenesis” section was remodeled. Parts 2.1, and 2.2. were removed from the text

Lines 87-104: The mechanism presented here is not described in the cited article No. 16 at the end of the paragraph.

The “Pathogenesis” section was remodeled. Parts 2.1, and 2.2. were removed from the text.

Lines 105-123: The whole paragraph (2.1. Sequestration - by the way I think it should be 2.2.) is based on one citation. Can the authors cite any studies on B. gibsoni associated with the mechanism described in this paragraph? E.g. One study showed increased concnetration of TNF in B. gibsoni infected dogs (https://doi.org/10.1016/j.exppara.2015.08.002)

The “Pathogenesis” section was remodeled. Parts 2.1, and 2.2. were removed from the text, article https://doi.org/10.1016/j.exppara.2015.08.002 was cited in the text.

Line 126: I think the anemia is regenerative. No word about degenerative anemia in cited article No. 18.

The Reviewer is right. In the course of canine babesiosis due to B. gibsoni, regenerative anemia is observed. This mistake was corrected in the text.

Lines 131-136: This mechanism has not been described in cited here article No. 19., and Lines 139-146: I cannot verify this paragraph. I have no access to the article No. 20.

These parts of the text were reformatted.

Reviewer 2 Report

The manuscript by Teodorowski is a well-written and concise review on babesia infections in dogs, from a European perspective. The authors describe babesia related pathogenesis and distribution in Europe. Overall, the manuscript is interesting but does require changes before publication in Animals. In addition to the narrow breadth of the review being a concern for this journal’s audience, below are a few major concerns to be addressed to help increase the clarity, accuracy and usefulness of this review.

  1. The manuscript is very poorly referenced for a review. In fact most sections run on for whole paragraphs, and only one reference was included. Please include additional references throughout the manuscript.
  2. References must be included within the table in Table 1
  3. The manuscript suffers from an inadequate use of illustrations. Please include schematics or figures that describe 1) animal pathogenesis and 2) European distribution.
  4. The authors should include a thorough discussion on Babesia diagnostics and recommendations.
  5. Please review and correct changes in font size.

Author Response

AUTHORS’ RESPONSES TO REVIEWER #2

We are very grateful to the Reviewer for his/her careful reading of the manuscript.

The manuscript by Teodorowski is a well-written and concise review on babesia infections in dogs, from a European perspective. The authors describe babesia related pathogenesis and distribution in Europe. Overall, the manuscript is interesting but does require changes before publication in Animals. In addition to the narrow breadth of the review being a concern for this journal’s audience, below are a few major concerns to be addressed to help increase the clarity, accuracy and usefulness of this review.

  1. The manuscript is very poorly referenced for a review. In fact most sections run on for whole paragraphs, and only one reference was included. Please include additional references throughout the manuscript.

We are very grateful to the Reviewer for his/her careful reading of the manuscript. We agree with all suggestion of the reviewer. The manuscript was reformatted. Section 2.1, and 2.2 were removed from the text. In a present form information presented in the text are focused mainly on B. gibsoni infection and proper citations were used in the text.

  1. References must be included within the table in Table 1

According to the suggestion of the Editor, Table 1 was removed from the text.

  1. The manuscript suffers from an inadequate use of illustrations. Please include schematics or figures that describe 1) animal pathogenesis and 2) European distribution.

As suggested by the Editor additionally figure presenting European distribution of B. gibsoni  was added to manuscript.

  1. The authors should include a thorough discussion on Babesia diagnostics and recommendations.

Although the diagnosis of Babesia gibsoni infection was not the scope of the paper, some information about diagnostic methods were added to Discussion section, as suggested by the Reviewer.

  1. Please review and correct changes in font size.

Changes in font size were corrected as suggested by the Reviewer.

Reviewer 3 Report

I have few comments to improve the manuscript.

1: Figure 1 is not so clear and missing of scale bar.
2: Figure 2 is need to show the distribution of Babesia gibsoni in European countries
3: Regarding this review is only for Babesia gibsoni, I'm confusing why showing so many information about other Babesia parasites in the table 1.
4: Haemaphysalis longicornis is a confirmed vector of Babesia gibsoni.
5: Treatment of Babesia gibsoni should not limited to azithromycin with atovaquone, clindamycin. Some many potential drug candidates indicating good result currently.
6: So, how do we face to the appearing Babesia gibsoni challenge in Europe?
7: Sequestration is 2.2? Blood stasis seems 2.1.

Author Response

AUTHORS’ RESPONSES TO REVIEWER #3

We are very grateful to the Reviewer for his/her careful reading of the manuscript.

I have few comments to improve the manuscript.

  1. Figure 1 is not so clear and missing of scale bar.

The quality of figure 1 was improved, and scale bar was added.

2: Figure 2 is need to show the distribution of Babesia gibsoni in European countries

Figure 2 with distribution of B. gibsoni in Europe was added to the text as suggested by the Reviewer.

3: Regarding this review is only for Babesia gibsoni, I'm confusing why showing so many information about other Babesia parasites in the table 1.

According to the suggestion of the Editor, Table 1 was removed from the text.

4: Haemaphysalis longicornis is a confirmed vector of Babesia gibsoni.

Information about possible vectors of B. gibsoni was added to the text.

5: Treatment of Babesia gibsoni should not limited to azithromycin with atovaquone, clindamycin. Some many potential drug candidates indicating good result currently.

Although the therapy of Babesia gibsoni infection was not the scope of the paper, some information about treatment were added to Discussion section, as suggested by the Reviewer.

6: So, how do we face to the appearing Babesia gibsoni challenge in Europe?

In conclusion, this infection should be suspected if the patient is an American Pit Bull Terrier or a related dog breed. It is expected that in the future, B. gibsoni infections may appear in other nonendemic regions in Europe, which may pose significant challenges for veterinary practitioners. Constant monitoring of B. gibsoni infection cases in dogs in Europe, as well as  and the creation of maps of their occurrence in individual countries will allow for an easier analysis of the epidemiological situation of this tick-borne disease and will contribute to its more effective diagnosis, therapy and prevention.

7: Sequestration is 2.2? Blood stasis seems 2.1.

This part of the manuscript was reformatted

Reviewer 4 Report

The manuscript “Babesia gibsoni Infection in Dogs – a European Perspective” by Teodorowski and colleagues provided a good review of published studies on Babesia gibsoni in dogs in European countries.

I really liked the writing style, which made the manuscript easy to understand. The authors relied on 46 bibliographic references to conduct this review. I did a quick search on NCBI Pubmed for the terms below, and found only these few results, which match perfectly with the low number of publications you found.

“Babesia gibsoni AND dogs” = 363 results

“Babesia gibsoni AND Europe AND dogs” = 37 results

These few studies made me question the first paragraph of your conclusion, which says: "The literature review presented here indicates that B. gibsoni infections in dogs in Europe are not frequent (…)". In addition to being not frequent (when compared to B. canis, for example), there is also a lack of studies directed at B. gibsoni. Another fact is that the European continent has 44 countries, and you only found reports in 8 countries. You can talk a little about it in your conclusions.

Another suggestion is to include a Table 2 on prevalence and technique used, like this:

Table 2 Prevalence of canine infection by Babesia gibsoni in Europe

Country/Region

Prevalence (%)

Technique

Comments

Reference

Spain

Spain (Barcelona)

Germany

Croatia

Italy

Romania

Serbia

Slovakia

Poland

With a table, the results described are more visual and easier to understand. You can add more columns if you want.

I only have a few more comments below.

Line 51: Giemsa-stained? Please write this information in the legend. Please include a scale-bar in the figure 1

Line 52: B. canis

Table 1: Babesia spp. and Theileria spp. – “spp.” not in italic

Babesia microti-like – “like” not in italic

Please do not separate words in the table (e.g., Amer-ica and Theil-eria)

Line 152: B. gibsoni

Author Response

AUTHORS’ RESPONSES TO REVIEWER #4

We are very grateful to the Reviewer for his/her careful reading of the manuscript.

The manuscript “Babesia gibsoni Infection in Dogs – a European Perspective” by Teodorowski and colleagues provided a good review of published studies on Babesia gibsoni in dogs in European countries. I really liked the writing style, which made the manuscript easy to understand. The authors relied on 46 bibliographic references to conduct this review. I did a quick search on NCBI Pubmed for the terms below, and found only these few results, which match perfectly with the low number of publications you found.

“Babesia gibsoni AND dogs” = 363 results

“Babesia gibsoni AND Europe AND dogs” = 37 results

These few studies made me question the first paragraph of your conclusion, which says: "The literature review presented here indicates that B. gibsoni infections in dogs in Europe are not frequent (…)". In addition to being not frequent (when compared to B. canis, for example), there is also a lack of studies directed at B. gibsoni. Another fact is that the European continent has 44 countries, and you only found reports in 8 countries. You can talk a little about it in your conclusions.

The conclusion section was reformatted.

Another suggestion is to include a Table 2 on prevalence and technique used, like this:

Table 2 Prevalence of canine infection by Babesia gibsoni in Europe

Country/Region

Prevalence (%)

Technique

Comments

Reference

Spain

Spain (Barcelona)

Germany

Croatia

Italy

Romania

Serbia

Slovakia

Poland

With a table, the results described are more visual and easier to understand. You can add more columns if you want.

The new table was added to the manuscript, as suggested by the Reviewer.

I only have a few more comments below.

Line 51: Giemsa-stained? Please write this information in the legend. Please include a scale-bar in the figure 1

The information was added to figure legend, as suggested by the Reviewer.

Line 52: B. canis

It was corrected as suggested by the Reviewer.

Table 1: Babesia spp. and Theileria spp. – “spp.” not in italic

Babesia microti-like – “like” not in italic

Please do not separate words in the table (e.g., Amer-ica and Theil-eria)

According to the suggestion of the Editor, Table 1 was removed from the text.

Line 152: B. gibsoni

Corrected as suggested by the Reviewer.

Round 2

Reviewer 2 Report

The authors have addressed all my comments adequately.

Author Response

AUTHORS’ RESPONSES TO REVIEWER #2

We are very grateful to the Reviewer for his/her careful reading and accepting of the manuscript.